# The 'Anthropocene Proposal': A Possible Quandary and A Work-Around

**Martin Bohle** [1,2,3,*] **and Nic Bilham** [2,4,5]

1   DG Research and Innovation, European Commission, 1149 Brussels, Belgium
2   International Association for Promoting Geoethics (IAPG), 00143 Rome, Italy; nic.bilham@gmail.com
3   Ronin Institute for Independent Scholarship, Montclair, NJ 07043, USA
4   Business School, University of Exeter Penryn Campus, Penryn TR10 9FE, UK
5   Camborne School of Mines, University of Exeter Penryn Campus, Penryn TR10 9FE, UK
*   Correspondence: martin.bohle@skynet.be

**Abstract:** The debates about naming the unfolding times of anthropogenic global change the 'Anthropocene' are ultimately debates about the 'human condition'. The proposal to amend the geological time scale by adding an 'Anthropocene' epoch (that is, the 'Anthropocene proposal' in its strict sense) is both an intra-geoscience debate about scientific sense-making and a debate about the societal context of the geosciences. This essay juxtaposes these debates, starting from three postulates: first, that the scientific methods of geological chronostratigraphy are applied rigorously; second, that anthropogenic global change is happening; and third, that the 'Anthropocene proposal' may be rejected if it does not meet the conditions required for its approval based on the rigorous application of the scientific methods of geological chronostratigraphy. These postulates are analysed through the lenses of the Cape Town Statement on Geoethics and the normative statements of the 'geoethical promise'. It is found that an ethical quandary would arise if the 'Anthropocene proposal' were to be rejected. Consequently, and given the societal contexts of the geosciences, it is explored whether distinguishing between the geological past (as demarcated according to current chronostratigraphic methodology) and contemporary geological–historical times (characterised somewhat differently) could offer a work-around to tackle the quandary.

**Keywords:** Anthropocene; societal geosciences; geoethical promise; ethical dilemma; geological time scale

## 1. Introduction

Human activity has altered the Earth system's dynamics [1]. Therefore, naming contemporary times the 'Holocene' may seem insufficient to characterise the current state of the Earth system. To add the epoch 'Anthropocene' to the geological time scale, in the first instance, is an 'intra-geoscience' proposal about amending the International Chronostratigraphic Chart [2]. However, to name the unfolding times of anthropogenic global change the 'Anthropocene' is also a systemic statement about the 'human condition'. When understood in that sense, the proposed re-naming of contemporary geological times is a paradigmatic shift in the perception of how to situate humans within nature (for examples, see [3–7]). That paradigmatic shift makes the proposal to amend the geological time scale an 'extra-geoscience' undertaking too, from which the chronostratigraphy of the Quaternary, and indeed "stratigraphy's epistemic authority over the Anthropocene" [8] (p. 1), cannot escape. Although mutually entangled, the two suggestions—to amend the geological time scale, and/or to name the unfolding times of anthropogenic global change the 'Anthropocene'—are not the same. Hence, they may need to be tackled in a different way, at least in part; this may be deemed necessary especially when considering the societal contexts of the geosciences.

The fact that the suffix '-cene' (in 'Anthropocene') has already become firmly associated with both these proposals may be unfortunate in terms of established geological naming conventions and their implications, in that it is always used by geologists to denote an epoch—a term which in turn corresponds to a specific hierarchical subdivision of the geological time scale. It might be appropriate, in the end, to distinguish practically between 'intra-science' naming conventions and extra-science connotations. In the public sphere, the name 'Anthropocene' has already become established. Making sense of the concept it denotes matters greatly, because it has been assimilated deeply in extra-geoscience contexts. Therefore, this essay is about sense-making in the given circumstances, including established naming conventions and practices. No alternative naming of contemporary geological times is proposed here—this is a matter for competent chronostratigraphers. Instead, part of the focus of the essay is to sketch a work-around to handle the paradigmatic shift that is implied by a widespread usage in extra-geoscience contexts of the term 'Anthropocene', in the possible instance that it is not adopted formally within geosciences.

For convenience, the formal proposal to amend the geological time scale by declaring the Holocene epoch to be over and adding a subsequent epoch (the Anthropocene) is referred to in the following as the 'GTS amendment' ('GTS' denoting the geological time scale). This terminology is adopted to distinguish it from the much wider meaning of the debate over calling present times the 'Anthropocene', within and beyond the geosciences. The latter denotes the enduring impact that humankind has on the Earth system's dynamics, irrespective of whether (or not) the 'GTS amendment' is eventually approved. This essay sketches the societal contexts of the geosciences to sharpen questions about how to handle the 'GTS amendment' proposal; however, much more could be said about the societal contexts of sciences, and respectively of geosciences [9,10].

For more than a decade, geosciences and many other disciplines of natural sciences, social sciences and humanities have debated the notion 'Anthropocene', the concepts behind it and the societal implications that it expresses [11–14]. The vigour of these debates indicates that a profound issue is at stake of whether, to use Hamilton and Grinevald's summary, we are witnessing the emergence of "a kind of hybrid Earth, of nature injected with human will, however responsibly or irresponsibly that will may have been exercised" [15] (p. 68). To a substantial degree, the debates about the meaning of the notion 'Anthropocene' are about the contemporary 'human condition'; that is, they are about how people live, collectively, and how to mutually situate the social and natural features of their lives. Consequently, discussing the 'GTS amendment' is a discussion firstly about the societal context of the geosciences [16,17], and secondly about adding a further epoch to the Quaternary on the International Chronostratigraphic Chart—the Anthropocene.

To focus on the specific subject of this special issue [18], this essay takes as its starting point three postulates or premises. Together, they constitute an intersection of scientific/scholarly findings and methodological norms, which this essay analyses. As will be discussed, a quandary may arise from the three postulates.

It is postulated, first, that the rigour of the scientific methods applied in chronostratigraphy, through which the geological time scale has been developed, shall not be compromised. Second, it is postulated that anthropogenic global change to the Earth system is happening, and indeed that "the Anthropocene for the first time gave birth to a universal 'Anthropos'" [19] (p. 118). Third, it is postulated that the 'GTS amendment' might be rejected by the scientific process/body that is empowered within the International Union of Geological Sciences (IUGS) to define the International Chronostratigraphic Chart if it were to conclude that the conditions for a new geological epoch had not been met—under these conditions, such a course of action would align with the first postulate.

The societal context of the geosciences is outlined in Section 2 to explore the circumstances that might emerge if the 'GTS amendment' were to be rejected by geoscientists. The outline builds on the spirit of the Cape Town Statement on Geoethics that situates geoethics as "a way of thinking and practising geosciences, within the wider context of the roles of geoscientists interacting with colleagues, society and the planet" [20,21]. Next, the 'geoethical promise' [22] is applied as an analytical tool

(Section 3). The 'geoethical promise' is a Hippocratic-like oath for geoscientists that is embedded into the Cape Town Statement on Geoethics. Drawing on analyses of the societal environments of the geosciences, the 'geoethical promise' presents nine normative statements. Given the three postulates and drawing on the 'geoethical promise', it is explored what a presumed acceptance or rejection of the proposed 'GTS amendment' would represent.

By way of framing this exploration, it is understood that the competent experts (geologists, and specifically chronostratigraphers and those working in closely allied specialisms) define what is meant by 'the methodological rigour that amending the geological time scale requires'. This implies building the geological case, forensically and methodically, based on the geological evidence (see, for example, the description of this process given by Wagreich and Draganits in the introduction to their discussion of anomalies in the geological record resulting from lead mining and smelting as potential stratigraphic markers for the base of an early Anthropocene [23]) . Likewise, it is for the competent experts to assess what margins of interpretation may be possible or how scientific methods may evolve. These kinds of methodological questions are at the core of the debates in geological stratigraphy about the 'GTS amendment' [2,24–26]. Consequently, chronostratigraphic methodologies are taken in this essay 'as given'. Thus, drawing on the first postulate of this essay, it is anticipated that the chronostratigraphers will do their scientific work thoroughly, notwithstanding that it is laden with societal constraints [7,8,27–30]. Likewise, it is taken as a background assumption for the purposes of this essay that discussing anthropogenic global change requires engagement with issues like how to account for its history, environmental justice, social justice and system change [31–34]. An analysis of the social contexts of geosciences, in a broader sense than this essay presents, must explore such issues to bridge the cultural distance between geosciences and social sciences (or humanities). Thus, the following section outlines only some perspectives of the multiple social contexts of the geosciences [10].

## 2. Societal Contexts of the Geosciences

Geosciences or Earth sciences are an amalgam of fundamental and applied research fields mainly within, but also beyond, natural sciences. They also include specific engineering disciplines and commercial undertakings on various scales, ranging from individual chartered experts to state-owned or multinational private corporations. Geology is among the core disciplines of geosciences. Stratigraphy is that part of the geological disciplines that concerns the composition and relative positions of rock layers. Chronostratigraphy is a sub-discipline that uses stratigraphy to provide a temporal framework for geological history up to present times. Within the international scientific dealings of geologists, through the work of bodies such as the International Commission on Stratigraphy (a subsidiary body of the IUGS), sits the responsibility to describe the geological time scale and to assess putative amendments to it [2]. The notion 'Anthropocene' did not first emerge within the geoscience community that is now charged with assessing the proposal to amend the geological time scale, but instead emerged among Earth system scientists [35]. The Earth system sciences go beyond geosciences or Earth sciences, reaching out to social sciences [36], and emerged after the middle of the last century.

Irrespective of how these research fields (or disciplines) are prescribed or distinguished, they aggregate natural-science knowledge about the functioning of the Earth, albeit with an initial focus on the abiotic components of the Earth systems, and they have limited interaction with social sciences or humanities beyond economics. Together, these geoscience disciplines nourish a corpus of stewardship knowledge about natural processes that can inform how people could act within the Earth system [37–40]. Contemporary geoscience knowledge is therefore of very high operational value for the functioning of modern societies.

That geoscience knowledge alone, however, does not guide how people ought to act. That issue is addressed by ethics in general, as well as in the specific form of professional ethics. However, even in the absence of guidance in how to act, a geoscientist's expert knowledge comes with responsibility for the individual scientist, as a professional and as a citizen, towards people and communities [20,21,41,42].

Consequently, the formal naming of contemporary times of anthropogenic global change in line with geoscientific notions is not an impartial act that is remote from societal considerations.

Unsurprisingly, the initial proposal to name contemporary times the 'Anthropocene' [35] triggered a range of reactions. Scholars in social sciences, philosophy and humanities have criticised it as concealing the responsibility of particular actors and the historical contexts that led to it [43–53]. Mutatis mutandis, the notion 'Anthropocene' emerged from these debates as a shorthand for our times.

### 2.1. Planetary Human Agency and Anthropogenic Global Change

During prehistoric and historical periods, humankind modified natural environments into appropriate resources for living and wellbeing [54–56]. Contemporary societies apply geosciences extensively for their economic, societal and cultural activities [50,57–60] and bind, through global supply chains, the entire globe into one social-ecological system [61] that intersects deeply with the physical and biological systems of the Earth.

Crafts-persons, technicians, architects and engineers apply geoscience knowledge, at least implicitly, when altering natural environments or creating artefacts (e.g., the extraction of minerals, the laying of foundations for buildings or managing floodplains). Artists, poets or philosophers of any time or culture connect with the Earth for shaping human identity. Contemporary geoscience knowledge seeps into modern thinking and dealings [62,63], often without being identified as such [64,65]. This sometimes blurs the interface between culture and nature [66,67], and has rarely been put forward so openly as in the metaphorical title of the book by the geochemists Langmuir and Broecker [68], *How to Build a Habitable Planet*.

Large-scale infrastructures like shore defences, hydropower plants or urban dwellings visibly interact with the geosphere and are a physical expression of how people situate themselves on Earth, views of which alter through history [69–71]. Whatever the philosophical concepts are that frame the construction of these infrastructures, they could not have been built without a profound geoscience culture [72–75] that includes scientific understanding, technological know-how and societal justifications. Likewise, purposefully designed global production systems or consumption patterns couple human activity with the geosphere at a planetary scale. The coupling happens through cycles of matter, energy and information [76–78] that are mostly invisible. Greenhouse gas emissions are well-known as the most prominent example, although a similar case could be made for nitrogen or the global agriculture system [79–81].

During the last century, humankind's activities have intersected with the geosphere in a much more extensive and intricate manner than ever before, either directly or intermediated through the biosphere [82,83]. Over some decades, the increasing number of people living on Earth and more notably the profligate consumption of resources in the affluent industrialised regions has culminated in anthropogenic change at a planetary scale [6,37,84].

When considering the outcome of recent economic activity and its cultural justifications (e.g., by hegemonic systems of cultural values), the resulting anthropogenic global change may be considered as being intentional, at least tacitly, because it is happening through negligence [31,70,85]. Consequently, it is appropriate to qualify anthropogenic global change as driven by an 'Anthropos' that is organised at a planetary scale; change in which, as in the classic Western movie, the good, the bad and the ugly are mutually entangled [86], and that is, likely, inescapable [87]. Notwithstanding such systemic human agency, individual (or collective) human beings of past and present times have not all caused anthropogenic change in the same manner or to the same extent [29,31,44,88,89]. Therefore, the notion 'Anthropocene' should be used while also acknowledging the responsibilities, political mechanisms and social processes that led to the current state of the globe. Nevertheless, anthropogenic global change is about how people, given hegemonic systems of cultural values, choices and lifestyles, govern the appropriation of biotic and abiotic resources from natural environments at a planetary scale [90]. This description of the 'human condition' would be the essence of a geological epoch named the

'Anthropocene', in light of which, proposing and handling the 'GTS amendment' is an act of responsible science [9,91–93] that geoscientists must face.

### 2.2. Planetary Human Agency and Geosciences

How societies alter natural environments depends on their technological means, the cultural views on how to deploy them, the scientific insights that underpin those technological means and cultural views and economic conditions, constraints and available resources. Together, they determine which 'endeavours' of anthropogenic change are possible or desirable to undertake. The principal human endeavour in contemporary times is to operate a 'technosphere' at the planetary scale [36,37,39,76,94–96]. In future, such a planetary technosphere may include deliberate scenarios of geoengineering the climate system [97–100]. Besides such scenarios, it is important to understand ongoing anthropogenic global change as driven by the cumulative effect of ordinary economic and social activities. That is an insight that justifies the notion 'Anthropocene' [50,78,101,102].

Within society's corpus of technological means, cultural views and scientific insights, geoscience knowledge has the potential to fundamentally shape the direction, effectiveness and efficiency of anthropogenic change of Earth system dynamics. To that end, when answering questions about the Earth system like 'where to situate humankind', 'how to change processes' or 'what features to safeguard', the geosciences provide 'instruments' such as Earth science literacy, insights into the origin of Earth (including its development through aeons) and an understanding of how Earth system dynamics operate.

The applied geoscience professions use established knowledge, methodologies and technologies to change the physical world [103]. Engineering geoscience disciplines pursue control of processes (that fundamental geoscience disciplines identify), interface applied and fundamental fields and link with other engineering disciplines to harness human power to alter Earth system dynamics. This amalgam of fundamental, applied and engineering sciences, as well as the commercial use of geoscience knowledge, makes geosciences an instrument to further anthropogenic global change. The working environments of applied and commercial geosciences explicitly frame the activities of 'geo-professionals' who are subject to ethical requirements and professional responsibilities that cascade through the links with applied and engineering disciplines, and that also engage, ultimately, with the geoscientists doing fundamental research [104].

When considering anthropogenic global change in its daily societal context, people make use of geoscience knowledge. This is because any given individual interacts with the Earth system, be it only as a consumer of resources. Furthermore, citizens need insights into the functioning of the Earth system to engage in better-informed decision-making [42]. The notion 'Anthropocene' would summarise such insights in the powerful message that the natural world and the human world are tightly interconnected and that, as a result, human activities have already irrevocably altered the Earth system [1]. Consequently, insights stemming from Earth system sciences about such matters as planetary boundaries [105] could inform citizens about 'what to do' and 'what not to do'. Hence, it is essential that geoscientists handle the scientific notion 'Anthropocene' responsibly, including the fate of the question of whether to amend the International Chronostratigraphic Chart formally. This responsibility results from the specific function that geoscientists have within contemporary societies because of the corpus of expertise that they can offer.

To summarise, geosciences are instrumental in making anthropogenic global change happen. Therefore, geoscientists are its co-architects who should assume the responsibility that comes with their role as agents of technology-driven change [10,92]. In this context, how geologists use their expertise is not an impartial matter; this includes what to do with the 'GTS amendment' proposal.

## 3. Ethical Contexts of the Geosciences and the 'GTS Amendment'

Science and research should be in service to society [9], and responsible science is a public good [93]. Furthermore, any undertaking of science and research is value-laden [106]. These insights have taken

root in contemporary societies [91] and, although they are still questioned to some degree, they have become operationalised [107]. Like many other natural science communities, the geosciences have strengthened their professional ethical frameworks in the last decade. Frequently, these frameworks, which consider the works of professional geoscientists in various societal contexts, use the label 'geoethics'. Mainstream geoethics is an example of an agent-centric virtue-ethics that puts the responsible behaviour of a human agent at the centre of thinking about human–geosphere interactions, in regards to sustainable and inclusive economic and social development [10,21,41,108,109].

A 'geoethical promise' was formulated some years ago to render geoethics operational in educational contexts [22]. It was updated in the Cape Town Statement on Geoethics [20], which was proposed on the occasion of the 35th International Geological Congress (2016) in Cape Town. Regarding the roles and responsibilities of geoscientists in the face of global change, the Cape Town Statement on Geoethics states: "Geoscientists have know-how that is essential to orientate societies towards more sustainable practices in our conscious interactions with the Earth system. Applying a wider knowledge-base than natural sciences, geoscientists need to take multidisciplinary approaches to economic and environmental problems, embracing (geo)ethical and social perspectives. Geoscientists are primarily at the service of society. This is the deeper purpose of their activity." [110].

The 'geoethical promise' offers nine normative statements (Table 1) that refer to the professional practices of geoscientists. These statements call for an ethical professional practice that recognises the need for high intra-professional standards as well as the obligations of geoscientists towards society and the Earth. Hence, the 'geoethical promise' offers geologists, and others [111], a framework to analyse implications of choices to be made in professional contexts. Other frameworks are available [112–115], but none is comparable to the 'geoethical promise' in its operational form. Therefore, this simple tool is used in the following discussion.

**Table 1.** Geoethical promise applied to the 'GTS amendment'.

| Statements Made in the 'Geoethical Promise' | . . . When Applied to the 'GTS Amendment' Proposal. |
|---|---|
| (I) . . . I will practice geosciences being fully aware of the societal implications, and I will do my best for the protection of the Earth system for the benefit of humankind. | . . . then these statements could be interpreted as calling for geoscientists (and others) to be aware of ongoing anthropogenic global change, giving this awareness the highest priority. Scientifically naming the contemporary times the 'Anthropocene' would raise awareness to promote sustainable development. |
| (II) . . . I understand my responsibilities towards society, future generations and the Earth for sustainable development. | |
| (III) . . . I will put the interest of society foremost in my work. | |
| (IV) . . . I will never misuse my geoscience knowledge, resisting constraint or coercion. | . . . then these statements call on geoscientists to be uncompromising vis-à-vis third-party requests regarding the application of geoscience knowledge and methodology. |
| (V) . . . I will always be ready to provide my professional assistance when needed, and I will be impartial in making my expertise available to decision makers. | |
| (VI) . . . I will continue the lifelong development of my geoscientific knowledge. | |
| (VII) . . . I will always maintain intellectual honesty in my work, being aware of the limits of my competencies and skills. | . . . then this statement calls for truthfulness in applying geoscience knowledge and methodology. |
| (VIII) . . . I will act to foster progress in the geosciences, the sharing of geoscientific knowledge, and the dissemination of the geoethical approach. | |
| (IX) . . . I will always be fully respectful of Earth processes in my work as a geoscientist. | |

*3.1. The 'GTS Amendment' seen through the Lens of the 'Geoethical Promise'*

The nine statements of the 'geoethical promise' (see Table 1) are used in the following to frame the 'GTS amendment' from a geoethical perspective.

When applied to the 'GTS amendment', some statements of the 'geoethical promise' could be interpreted as calling on geoscientists to make people aware of ongoing anthropogenic global change and to give this message importance. Three further statements could be interpreted as calling on

geoscientists to be uncompromising vis-à-vis third-party requests regarding how geoscience knowledge and methodology should be applied. Statements VI, VIII and IX seem not to offer any insight into how to appreciate the 'GTS amendment'.

Statements I, II and III of the 'geoethical promise' emphasise the societal responsibility of geoscientists: protection of the Earth for the benefit of humanity and future generations, sustainable development and acting in the interest of society are matters of concern that are of high relevance to the meaning of the notion 'Anthropocene'. Since anthropogenic global change is happening and threatens the future living conditions of people, these are matters of concern for any geoscientist. Therefore, geoscientists should make people, including individual or collective human agents with decision-making powers, aware of the threats that ongoing anthropogenic global change implies. Naming the present geological times the 'Anthropocene' would be an explicit message telling everybody about the size and nature of the ongoing change that humans are driving. Thus, it follows that the 'GTS amendment' should be looked upon favourably because of its societal relevance.

Statements IV, V and VII of the 'geoethical promise' are reflections about the societal status of 'scientific methods'. The wording of the statements is generic and applies to all scientific and scholarly activities. The wording does not refer to the specific professional or societal contexts of geosciences. To put it simply, the 'geoethical promise' calls for a methodological rigour that does not compromise according to (societal) pressures of any nature. Resistance to constraints, impartiality and honesty are matters of general concern. They are of paramount importance for the soundness of science, research and scholarship in any discipline. These statements, to put it differently, call for scientific methodology to be sheltered from requests to make them conform with needs that are external to the scientific inquiry that is being pursued. This imperative of methodological rigour is a result of a centuries-old process that has shaped scientific methods and should be key to any scientific endeavour. Thus, it is derived that the 'GTS amendment' should be looked upon favourably only if it fits the scientific methodology that is used to define the geological time scale.

### 3.2. Accepting or Rejecting the 'GTS Amendment'—An Ethical Matter

Summarising the investigation undertaken so far:

- Geoscientists have a particular societal responsibility because of the corpus of expertise that they can offer in times of anthropogenic global change, including what to do with the 'GTS amendment' proposal.
- Geoscientists have debated for some time whether and how a 'GTS amendment' may be accommodated within the geological time scale. Respecting the methodological rigour of the scientific methods of chronostratigraphy is a crucial issue of these debates (for examples see [11,116,117]).
- Within the context of debates about scientific methodology, the competent experts (geologists including chronostratigraphers) define what 'methodological rigour in amending the geological time scale' requires or how scientific methods to describe the geological time scale may evolve. Within that constraint, chronostratigraphic methodologies are taken 'as given'.

Drawing on the analysis in Section 3.1, two possible outcomes are considered:

- The first is that the bodies of the IUGS agree to the 'GTS amendment' having also upheld the condition that the proposal meets the existing methodology to determine the International Chronostratigraphic Chart (in line with the first of the three postulates set out earlier in this essay). In this case, all six relevant statements of the 'geoethical promise' (I–III, IV, V and VII) would be complied with. It follows that, when seen through the lens of the 'geoethical promise', the decision to approve the 'GTS amendment' would not give rise to an ethical issue, and no further analysis would be deemed necessary.
- The second is that the bodies of the IUGS uphold the methodological rigour required for amending the International Chronostratigraphic Chart but reject the 'GTS amendment' because it does not

meet the existing scientific methodology to determine the International Chronostratigraphic Chart. In this case, the guidance of the 'geoethical promise' would be split. Statements IV, V and VII (about impartiality and 'scientific methods') would be complied with, but statements I–III (about the societal context of the geosciences) would not be complied with. It follows that, when seen through the lens of the 'geoethical promise', the decision to reject the 'GTS amendment' would give rise to an ethical quandary, and further analysis would be needed.

The logic of this analysis is summarised in Figure 1.

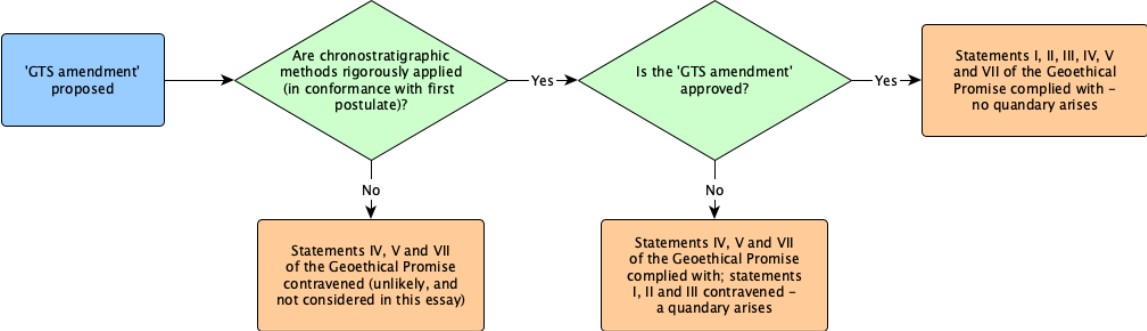

**Figure 1.** Possible outcomes and geoethical consequences of the 'GTS amendment' proposal.

## 4. Discussion

Within the analysis undertaken above, the first case, acceptance of the 'GTS amendment', would not cause ethical issues that in turn would require further analysis. The contrary applies to the second case, in which an ethical quandary would emerge. In the case of split guidance by the statements of the 'geoethical promise', as in the second case, the 'geoethical promise' itself does not offer counsel on how the arising difficulty should be handled. Thus, the ethical quandary that would arise in the second case cannot be handled within the analytical framework that has been applied so far; this complicates the subsequent analysis.

### 4.1. What Ethical Quandary would Arise from Rejecting the 'GTS Amendment'?

As derived above, under the premises of this essay: In the (possible) circumstance that the 'GTS amendment' is rejected because of failing the methodological requirements for setting the geological time scale, applying the 'geoethical promise' would lead to an ethical quandary. Two ethically essential considerations (cultural values) would thus be pitched against each other.

On one side, given the societal responsibility that they possess, geoscientists must inform society about the nature of contemporary times in an appropriate manner. The requirement to use scientific findings to improve human societies (meaning to inform people about anthropogenic global change, in the given case) is a cultural value underpinning why science is undertaken in the first place. Hence, this value constitutes guidance as to what scientists ought to do. On the other side is a requirement of the scientific process that scientists should not adjust scientific methodology in response to an external appeal. The rigour of scientific methods is an outstanding cultural value that required centuries to be established. Hence, this value constitutes guidance as to what scientists ought to do. Under the assumptions made in this essay, attempting to comply with both values would create an ethical dilemma in the instance that the 'GTS amendment' is rejected.

Although 'informing society about the nature of contemporary times in an appropriate manner' could be done by other means than scientifically calling them the 'Anthropocene', the vigour of debates that this notion has triggered witnesses the power of the message that comes with that naming. Notwithstanding extra-geoscience debates, saying that the *Anthropocene is Functionally and Stratigraphically Distinct from the Holocene*, as the title of one of the publications of the proponents of the 'GTS amendment' [1] does, sends a very determined message, particularly when it comes from the

geoscience community. Such a message would be of very high societal relevance and value if embraced by the discipline and formally endorsed by the scientific body empowered to do so. It would have the potential to orientate citizens and decision makers. Hence, the question arises of what to do if the 'GTS amendment' is judged to be methodically flawed when it is assessed from the perspective of scientific methodology for determining the geological time scale, given that anthropogenic global change is ongoing and that citizens should be kept informed.

The 'geoethical promise' does not offer a remedy in case of an ethical dilemma or quandary of conflicting ethical values. Conceptually, to handle this challenge, one might contemplate a hierarchy of the normative statements of the 'geoethical promise' drawing on other pre-existing value systems. For instance, one might apply Jonas' [92] concept of an 'imperative of responsibility' to deduce that the statements I to III of the 'geoethical promise' should be given greater weight than statements IV, V and VII. Such an adjustment of the 'geoethical promise' would accentuate the ethical dilemma if the 'GTS amendment' were to be rejected on methodological grounds, but would in turn offer a means to overrule the obligation to comply with statements IV, V and VII (about impartiality and scientific methods) of the 'geoethical promise'. Alternatively, one might apply a utilitarian approach, drawing on existing work on scientific methodology to argue for a greater weight for statements IV, V and VII of the 'geoethical promise' than statements I to III, a choice that would safeguard scientific methods as a paramount cultural achievement; consequently, statements I, II and III (about the societal context and value of the geosciences) would be overruled. None of these suggestions are taken up here, because in the context of the present analysis the 'geoethical promise' is the chosen ethical framework for analysis. It should be used with methodological rigour; namely, 'as-is', and without amending it. Hence, a revision of the 'geoethical promise' is not an option that is pursued here to handle the given quandary. Furthermore, it is far from clear that modifications to the 'geoethical promise', such as those mentioned above, would be a desirable undertaking. Any attempt to privilege either the social value of geoscience or its methodological rigour over the other in a generic sense could be profoundly damaging, either to the reliability of geoscience research and practice, or to its social efficacy, or to both. It may be neither possible nor desirable to design generic ethical guidelines so as to eliminate true ethical dilemmas—instead, it may be preferable to address such dilemmas openly and honestly, engaging both geoscientists and other citizens.

Notwithstanding the particularities of a chosen scientific methodology, experience teaches that when the application of a given methodology results in an ambiguous interpretation, then a remedy may be found 'outside' that framework, or, as a philosopher of the 19th century wrote, "The philosophers have only interpreted the world . . . " [118] (p. 533). The geological time scale (as set out in the International Chronostratigraphic Chart) is an interpretation of the stratigraphic record, albeit using scientific methods. That interpretation includes at least one specific feature that has no specific geological meaning, the 'zero-point' from which the years of the past are counted, wherever it is placed in the recent past [119]. That feature may provide a stepping-stone to seek a remedy to the quandary.

### 4.2. What Remedy may be Found?

On one side, the development of the geological time scale, including the methods used to establish it, is an essential cultural achievement [120]. On the other side, a scientifically correct and publicly meaningful identification of the current times in regards to anthropogenic global change would likewise be an essential cultural achievement. Scientifically assigning these times a name such as the 'Anthropocene' has paramount cultural value because it describes a paradigm shift in humankind's perception of its place in the world [16]. Both cultural achievements are essential for sound scientific practice, and it would be an unfortunate outcome to lose one of them in favour of the other.

It is important to acknowledge these cultural achievements. It is also important to distinguish between the temporal frameworks of, on the one hand, the geological past, and on the other, contemporary Earth system dynamics. Drawing on these insights, it seems plausible that a chronostratigraphic interpretation of the geological past (as represented on the geological time

scale) may not provide an entirely appropriate framework to apply to present and recent historical times of rapid anthropogenic global change [30]. Currently, the Holocene "is the name given to the most recent interval of Earth history, which extends to and *includes the present day*" [119] (pp. 3 and 4; emphasis in the original). The intention of that statement is evident although the interpretation that it offers is not uncontested [1].

Geologists developed the geological time scale (International Chronostratigraphic Chart) to describe a distant past of millions or billions of years. Although the geological time scale includes the present, it is mainly about the history of Earth in deep time, driven by geological processes. These processes continue; rocks are formed, deformed and eroded. Seen at that geological timescale, human history is an almost instantaneous event, more like a point than a period of time. Therefore, it seems appropriate to query the undifferentiated juxtaposition of these widely differing time frames within one analytical framework [34,121], although such an enormous difference in time scales does not apply when the duration of the Holocene is compared with the extent of human history.

In light of these considerations, and building on the foregoing analysis, this essay ends its exploration by considering whether a distinction might usefully be made between the geological past and contemporary geological times (that is, the present day and recent historical times). It is suggested that adding a 'transition point' to the International Chronostratigarphic Chart be considered. The 'transition point' would be is situated in the recent past of human history. It would mark the beginning of 'geologically contemporary' times. It would recognise that what comes after that point is qualitatively different to what preceded it. A 'transition point' of this sort would also allow for the possibility that a single, undifferentiated methodological framework (with a single hierarchy of time periods and corresponding geological units) cannot adequately characterise both the geological past and the 'geologically contemporary'. The qualitative differences between the geological past and the 'geologically contemporary' reflect the effects of an 'additional agent'—namely, humans. A carefully considered shift in chronostratigraphic methodology could occur at the 'transition point'. Such a shift could be designed with scientific rigour, and incorporated within a single International Chronostratigarphic Chart encompassing both the geological past and the 'geologically contemporary'.

This proposal seems attractive, therefore, from the point of view of developing an appropriate framework for characterising and understanding 'geologically contemporary' (human-influenced) times. As discussed below, it also provides a 'work-around' for the possible quandary of having to handle conflicting cultural values that would otherwise be caused by a rejection of the 'GTS amendment', as outlined above. How might such a proposal be implemented?

The International Chronostratigraphic Chart currently refers to a year in the recent historical past (a kind of 'zero-point') from which the years of the geological past are counted, as referred to, for example, in [119]: "an age of 11.700 calendar yr b2k (before AD 2000) for the base of the Holocene, with a maximum counting error of 99 yr" (p. 1). This kind of reference year is a matter of convention, chosen for convenience. It does not have any particular geological meaning. However, those who have the competence and authority to define and amend the geological time scale could give a meaning to such a 'zero-point' date, for example, by choosing it as the 'transition point' at which the geological past ends and the 'geologically contemporary' starts. It would denote that a profound change has taken place that should be recognised. The interference of human activities with the dynamics of the Earth system at a global scale would constitute such a profound change. The naming convention for the demarcation of time after the 'transition point' would remain to be determined. The term 'Anthropocene' would be one option. However, the chronostratigraphers may decide that a term other than 'Anthropocene' should be used to denote contemporary geological times (post-transition-point in the GTS), given the existing stratigraphic and geochronological implications of the suffix '-cene', and changes to naming conventions could legitimately be defined and adopted within the geological community. That said, the term 'Anthropocene' is already extensively used in a wider non-geoscience sense across a variety of scholarly and public settings. Rolling back these wider usages may not be a realistic ambition, given that this is not a matter entirely within the control of chronostratigraphers [8] (p. 4).

Diverse options would be available for establishing a date for the transition from the geological past to the 'geologically contemporary', drawing on existing and ongoing research into potential markers for the onset of the Anthropocene [23,25,122–125]. The peak of the plutonium fallout from nuclear tests in the atmosphere in the early sixties of the last century might be a candidate [117,126]. The 'Orbis spike' in the early seventeenth century might be an alternative [127] for a 'transition point' of the geological time scale. Debates within geoscience communities could move on from what may mark the transition from the Holocene to an Anthropocene, and instead consider the more engaging and societally relevant question of what the geological or geochemical markers of the (profoundly human-influenced) geological present are.

In that context (and going beyond 'geology-only' considerations), it may be fruitful to explore together with the stratigraphers of human history—archaeologists—when geological stratigraphy and archaeological stratigraphy could be regarded as meeting or overlapping in the geomorphological record [23,128–132]. Accordingly, one might set the 'transition point' of the International Chronostratigarphic Chart at that horizon. Consequently, one could consider more-recent times as belonging to the 'geologically contemporary', and either simply name it as such or give it a scientific name that seems suitable for such an amended geological time scale.

### 4.3. What Reconciliations Can Be Achieved?

Setting a 'transition point' within the geological time scale would circumvent the ethical dilemma that rejecting the 'GTS amendment' might cause, as perceived through the lens of the 'geoethical promise'. Naming that 'transition point' and what comes after it appropriately would be a practical work-around to limit possible damage to the societal function of the geosciences, because it would send a strong, public-facing message that human activities are profoundly affecting the Earth system's dynamics, and that this change is inscribed in the geological record. In doing so, the geological community would be providing a fundamental service to society. At the same time, a single geological time scale, including both the geological past and the 'geologically contemporary' meeting at a 'transition point', would reflect that rocks continue to be formed, deformed and eroded by geological processes, irrespective of what role human activities play in addition.

As mentioned already, alternative remedies of the quandary may be possible. For example, a hierarchy of ethical values could be developed for the societal context of the geosciences so that an accordingly reworked 'geoethical promise' would not lead to the identified ethical dilemma when applied to the 'GTS amendment'. It is, after all, a relatively new and untested tool for the analysis of geoethical challenges. However, the ethical dilemma it points out in this instance seems to be a real one, reflecting a genuine tension between two fundamental values of geoscience, irrespective of their particular formulation in the 'geoethical promise'. Hence, whatever its imperfections and shortcomings might be, it would be difficult to justify amending the 'geoethical promise', having selected it as the analytical framework for this essay, simply to avoid this dilemma and any others like it. Furthermore, to do so would be a complex inquiry of uncertain outcome and would require the development and widespread acceptance of rigorous methods and criteria, just as is the case when seeking to identify geological/geochemical markers of the geological past and the 'geologically contemporary'.

Finally, another remedy might be merely to stall the current debate about whether to extend the International Chronostratigraphic Chart by adding the 'Anthropocene', whether just for a few decades or a couple of centuries, to allow the dust to settle a bit, so to speak. It may be easier to discern whether and exactly when we have entered a new geological epoch with the benefit of more hindsight. However, considering the vigour of the ongoing debates around the notion of the 'Anthropocene', it seems unlikely that such a moratorium would be accepted. Furthermore, in doing so, the geoscience community would miss a unique opportunity to assume firmly their societal responsibility. As discussed above, this opportunity would also be jeopardised in the eventuality that the 'GTS amendment' proposal continues to be debated and is then rejected. On the other hand, to declare a 'transition point' within the geological time scale would constitute an active decision on the

part of the geoscience community to embrace its societal responsibility. For what stronger statement could there be of the geological community's verdict on human–geosphere interactions than to say that, although geological processes continue to operate, human activity has effectively brought geological history, as we have recognised and characterised it so far, to a halt?

## 5. Conclusions

Questioning the possible fate (acceptance or rejection) of the proposal to amend the geological time scale by adding an 'Anthropocene' epoch (the 'GTS amendment') indicates the possibility of an ethical problem. When analysing the 'GTS amendment' through the lenses of the Cape Town Statement on Geoethics and the 'geoethical promise', two important cultural achievements may be found to be in conflict. The potential conflict would be between, on one hand, giving contemporary geological times a societally resonant name (e.g., 'Anthropocene'), and, on the other, maintaining the scientific rigour of the methods used to determine the geological time scale. However, given the societal relevance of the geosciences, only together can these two cultural achievements constitute sound societal and scientific practice. Hence, choosing between them would be invidious and highly undesirable. In the circumstance of the rejection of the 'GTS amendment', that quandary would arise and would eventually negatively affect the societal function of the geosciences.

When seeking options regarding how to remedy this quandary, it appears practical to distinguish in the geological time scale between the ´geological past' and the 'geologically contemporary', and to denote a transition point between them, situated in the (recent) past of human history. Within such a work-around, geoscientists might use somewhat different methodologies to describe and demarcate the geological past and the 'geologically contemporary' (that is, the present and the recent historical past). Subsequently, an appropriate naming of the current geological times could be promoted with scientific rigour as justified by its societal relevance [92,133] but without compromising the methodological rigour that underpins the setting of the geological time scale.

In terms of the geological naming convention for suffixes including '-cene', the term 'Anthropocene' may pose a problem, at least within geological sciences, if it is used for the 'geologically contemporary' without referring to a geological epoch. However, the term seems to be established as a cross-disciplinary shorthand for our times, and geoscientists can do little about it [8]. As observed by Lorimer: "Regardless of what the International Commission on Stratigraphy decides, the genie is out of the bottle" [29] (p. 123). Hence, accepting these circumstances, the message from the geoscience community in naming present times of the Earth system after the 'Anthropos' would be unequivocal. It would imply the maintenance of epistemic authority by recognising (with rigorous methodology) the (additional) agent whose collective actions along a distinct historical path (notwithstanding the responsibilities of different individuals and groups of people over time) have made the difference between the geological past and the 'geologically contemporary'. If that choice is deemed unsuitable, then it may be practical to locate a 'transition point' from the geological past to the contemporary, and to name that transition point appropriately using the Greek words for human and time ('Anthropos' and 'Kairos' or, alternatively, 'Chronos') [120].

Finally, taking either a philosophical or a utilitarian view, the cultural value of the geological time scale as well as scientific notions like the 'Anthropocene' make sense only if a human society exists. The 'GTS amendment' represents a statement that humankind has entered a stretch of time that should be named by referring to human agency. Consequently, making that statement also means that the stretch of time of the purely geological past has ended. Taking a futurist standpoint, one could observe that, having as a species achieved the strength to (brutally) tamper with Earth system behaviour, that capability is unlikely to fade away, although it may get refined. Hence, the 'Anthropocene' likely will be a lasting geological time division that humankind will witness, and about which we, as good stewards, should care.

**Author Contributions:** The work of the authors, Martin Bohle (M.B.) and Nic Bilham (N.B.), was shared as follows ('n.a'. is used for non-applicable taxonomy): conceptualization, M.B.; methodology, M.B.; software, n.a.; validation, N.B.; formal analysis, M.B.; investigation, n.a.; resources, n.a.; data curation, n.a.; writing—original draft preparation, M.B.; writing—review and editing, N.B. and M.B.; visualization, n.a.; supervision, n.a.; project administration, n.a.; funding acquisition, n.a.

**Funding:** This research received no external funding.

**Acknowledgments:** The lead author thanks his employer for the consent to cooperate with the International Association for Promoting Geoethics (IAPG) and to affiliate with the Ronin Institute. All views expressed herein are entire of the authors, do not reflect the position of the European institutions or bodies and do not in any way engage any of them. The authors thank MDPI reviewers and editors for their various suggestions on how to improve the contribution, for this as well as its earlier versions.

**Conflicts of Interest:** The authors declare no conflict of interest.

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
