# Peer review of "The ‘Anthropocene Proposal’: A Possible Quandary and A Work-Around"

_quaternary, doi:10.3390/quat2020019_

Round 1

Reviewer 1 Report

This is more a dissertation than a classic scientific article, so I have no suggestions to change/improve the manuscript which seems to me correct and interesting in the present form. 

The treated topic is consistent with the current scientific debate, but also arises ethical and philosophical  issues.

Author Response

- see attached file -

Reviewer 2 Report

This essay makes the case for the naming of an "Anthropocene" epoch using an ethical lens. The authors perceive two competing values: (1) they accept that humans have interfered significantly with the Earth system which calls for a separate "Anthropocene" and (2) they argue that scientific reasons should prevail for naming a new epoch within the Chronostratigraphic Chart by the Stratigraphic Commission. The authors recognize that an important reason exists in favour of naming a new epoch, namely the necessity for geoscientists to aid society in solving geoscience issues. To investigate this aspect they turn to the Cape Town declaration and discuss how key points of this declaration may sway the decision for or against the naming. However, if the Stratigraphic Commission is unwilling to declare a new time they offer a way out by making the argument for separating a "geologic present" from the geologic past.

The essay is interesting and timely. I find the argumentation sound. Because it is not a classic science paper it is not straightforward to review it under the given guidelines. However, in my opinion it provides for a compelling read that would be interesting not just for geoscientists but quite possibly for a wider audience. I have four suggestions:

1.  It may help to briefly provide some information about what criteria are typically applied to name a geologic epoch, especially how the base of a new geologic time is defined. This would make the article easier accessible to non-geoscientists (and would also help those of us not doing stratigraphy) because it would demonstrate the scientific rigour that goes into such a decision.

2.  I am confused by the title and some of the text, because I don't quite understand that there are three distinct "premises" that are coming together. In my understanding this is a scientific debate within a societal context, and the authors are making a case that a geoethical perspective can aid in coming to a scientific consensus that is relatable to society. I find it pre-emptive to assume that the stratigraphic commission will decide against a new epoch. I think the paper would not become weaker by pursuing both possibilities.

3.  I find the notion of a "geologic present" that may have lasted centuries difficult to grasp. Geology is about processes; although the stratigraphic timescale tries to pinpoint boundaries by finding the first documented occurrence of a certain process though I think nobody would expect that this corresponds to the blink of an eye (but probably that would be an additional argument to make). The argument in line 490f is countered by line 459; let alone the hubris that is implicit. I would suggest a slight rewording of this paragraph.

4.  I am wondering if the authors have considered adding a figure (for example, a concept map or flow chart) that summarizes their argumentation and would aid the reader. I think that more geoscientists should consider ethical issues, and such a figure may draw someone to read the article and would also help non-scientific audience (which I think could be a target audience).

Overall I find the essay well structured. The introduction (section 1) provides insights into the issue and outlines the structure of the paper. The second section makes clear why the recognition of the "GTS amendment" is congruent with the responsibility of geoscientists to help society. The third section considers key points of the Geoethical Promise, how these may argue for and against the "GTS amendment" and asks the question what if it were rejected. The discussion provides a way towards the "GTS amendment"; and the conclusions reiterate key arguments. I am impressed by the wealth of references the authors include to assist interested readers in finding more details about aspects of their argumentation.

 I do commend the authors for taking the time to thoroughly investigate this topic. I do recommend publication of this article after the authors and editorial board have considered suggestions by me and the other reviewers.

Author Response

- see attached file -

This manuscript is a resubmission of an earlier submission. The following is a list of the peer review reports and author responses from that submission.

Round 1

Reviewer 1 Report

see attached file

Reviewer 2 Report

See attached

Reviewer 3 Report

The argument is stimulating, interesting and novel despite the fact that much has been written on this perceived conflict of definitions - and the likely rejection of the Anthropocene proposal by the Startigraphic Commission.

The biggest issue here really is language. The grammar is poor and occassionally unusual words are chosen.

In addition, the way the 'Anthropos' is framed sidelines political implications of a generic and collective definition of humankind - this seriously glosses over the uneven distribution of past and current carbon load contributions, for instance, and the uneven distribution of the burden of future loss. As has been pointed out by historians and anthroipologists in particular, the issue of climate change CANNOT be meaningfully disentangled from issues of system change, from capitalism and social justice. It's alright to the bring the geoethical promise into play here, but the way that this is mostly presented within the geoscience community actually seems rather uninformed on those matters. It would be really good for future work at least to engage more seriously and more broadly with SSH scholarship on this matter - and perhaps for resulting papers to be co-authored.

This article, as it stands now, may have an impact on geoscientists but I predict it will have a hard time reaching out to other scientific communities.

Author Response

- see attached file -

Reviewer 4 Report

I think this is a highly original argument that may offer a way through the various stresses and fractures in the debate over the Anthropocene, especially recognotion of the fact that humans have shifted the functioning of the Earth System as against the discomfort of more traditional stratigraphers who don't like seeing their discipline move outside tradition boundaries. Also, there is an ethicical commitment embedded in the new notion of the Anthropocene that is hard to avoid. So I am very attracted to the idea of ruling a line over the top of the GTS and declaring that the non-human division ended in (say) 1950. The author does not hint at what is to follow. I'm sure he's thought about it but has decided to save it for later, but perhaps a sentence or two would be valuable somewhere in the paper.

Some more specific comments by  line number:

42-4 Yes, this poses the poblem very well

57 "global anthropogenic chnage"; need to add "to the Earth System" to make it clear we are not talking about, eg, forest clearing 2000 years ago

58-9 italicised sentence.  Need to explain what this means in context of the argument of this paper

65 first mention of the "geoethical promise"; need to explain what this is

79 "to reduce the cultural difference between the geosciences and the social sciences"; why does it have to be reduced?

85-6 might reference Latour here

93-4 Earth system sciences "include social sciences". I don't think they do, except geography and human ecology, which cross the boundary

149 "it conceals responsibilities etc". Does it? People say it does, but everytime they say it, it shows that it doesn't. This is a fuphy.

155 section 2.2 a really excellent argument about how geoescientists cannot escape their social responsibility, no matter how "technical" and detached they think they are being

161-2 Yes, one or two do think it's a conspiracy to permit geoengineering. But it's a silly idea and probably not worth giving oxygen to.

198 "Science and research are a service to society". They can also be a disservice.

209 I did not know about the Cape Town Statement, and it comes as a surprise. Since it frames the core argument I wonder whether it could be mentioned up front, including the abstract.

225- Very impressed with the way the author uses the Anthropocene idea to identify contradictions in the geoethical statement. Fascinating

293-96  This seems to me to be the central statement of the paper.

306-7 The Marx quote is a bit elliptical in this context

328 Peak plutonium fall-out in the early 1950s. I thought is was the early 1960s. Check.

334-5 "methodologically sound alternative, such as the Orbis spike". Zalasiewicz et al., responding to the original Orbis paper, argue that it is not methodologically sound because it doesn't understand the vital criterion of a detectable change in the functioning of the Earth system

339 "Some philosophers may dislike it". Why would they? And how many "philosophers" have anything to say at all about the Anthropocene?

353-5 Perfect ending sentence.

Round 2

Reviewer 2 Report

The revision provided by Martin Bohle to the manuscript “Failure at a triple-point, the ‘Anthropocene proposal’?” has not addressed virtually any of the comments I raised to this ms in my former review. Changing the title to “The ‘Anthropocene proposal’ – failing at a triple-point? is quite illustrative of the sort of revision done, in short, changing things so everything stays the same. Rewording here and there and moving sentences around the text is not revising a text at least not what is usually understood by a scientific revision. So I am not going to repeat the critical points already raised in my first review. The note by the author to the reviewers and editor is not even a rebuttal of my criticisms since it not addresses the fundamental questions I raised.

First, if the author pretends to understand the topic of the special issue literally, then, there is no need of this triple-point assumption. That is, if the author is not going to deal with the reasons upon which rejection of the Anthropocene Working Group proposal is going to be based is better to simply assume such a rejection and abandon all this vacuous triple-point rhetoric. Second, the author builds up an artificious and fallacious ethical dilemma that has nothing to do with the proposal by the Anthropocene Working Group as he inadvertently acknowledges. On one hand, the author states that the geoethical promise offers guidance if the AWG proposal is accepted and, on the other hand, he states that the geoethical promise has nothing to say if the proposal is rejected (¿? First paragraph of section 3.2). To be clear, the geoethical promise does not offer guidance for anything. It is simply a bunch of vacuous statements at some intermediate point between self-help discourses and the tables of the law that does not resist a minimum confrontation with real life. The Christian and charity-like inspiration of this geoethics is quite clear with reference to Jonas ‘imperative of responsibility’ and his Gnosticism: a sort of secret mystics for salvation. To be honest, this ms is better suited for L’Osservatore Romano than for a scientific journal. As I stated already it is neither scientific nor rigorous but just a vacuous divagation around this geoethics. To begin with, if the author pretends to build up any kind of rigorous, robust and scientific discussion about whatever ethics he has to refer to the materialist foundations of any ethics based on the teleological character of labour. To do this he has to refer to the research done by philosophers like Gÿorgy Lukács, if not to Spinoza, and from then on to philosophers of the Budapest school and followers (Mészáros, Infranca, Tertulian, etc.). Once the universal determinations of ethics have been established on such basis the historical determinations giving rise to different historical forms of ethics can be investigated. Pretending to settle some sort of universal ethic rules out of any kind of historical determination is simply a fallacy.

Conflating expressions such as ‘hegemonic value’ with ‘neoliberal consumer/market capitalism’ and with Sergio Leone’s movie does not have any sense (lines 161-163). What is hegemonic value? Is it related to Marx’s theory of value? Does he know what he is talking about? Neoliberal consumer/market capitalism is not even something one can recall from neoclassical economics or Keynesian economics, not to say Marxian economics. The author insists on manipulating Marx when he is just in the antipodes of pretending to the change something (lines 347-352). The entire ms is indeed a vacuous exercise for not to change anything. Please, leave Marx in peace.

Reviewer 3 Report

The thorough revision is appreciated.

Reviewer 4 Report

Revised paper. 

The revisions meet my suggested changes.

Round 3

Reviewer 2 Report

Hence, the criticism of the form and extent of the revision of the paper must be refuted.

A formal revision is not necessarily a scientific revision, in particular, when the review provided is critic regarding the main corpus of the ms, that is, the scientific foundations of geoethics and this question is not addressed in the revision.

Evidently, if the reviewer had chosen to write a paper on this topic, they would have taken a different approach to mine – that would be their prerogative, of course, but it does not mean that other approaches are invalid.

It is author’s duty to demonstrate that his approach is valid. It would be valid if it were scientifically sounded, which is the kind of approach supposed to be followed in a scientific journal, but this is not the case of the present ms. Many papers dealing with Anthropocene issues such as planetary boundaries, great acceleration, global change, chronostratigraphic units and boundaries, etc. - not only from the fields of natural sciences but also from social sciences - have contributed to build up an empirically-based theoretical understanding of the Anthropocene, which, by the way, is entirely materialist-based but certainly not Marxist. Many contributions undertake an analysis of the socioeconomic reasons underlying the Anthropocene too. They reach to conclusions that can be criticized or not but at least they are based on concrete and material evidences, on some kind of scientific approach, and they are not a list of moral principles about how to be good people in the Anthropocene.

The discussions within geosciences (about the proposal to amend the Geological Time Scale) are about methods. They are not about political / societal sense-making of the proposal to name the present times ‘Anthropocene’

This is wrong. For example - and there are other examples - discussions about the lower limit of the Anthropocene are both methodological and political as many researchers from the fields of social and natural sciences have pointed out. Besides, it could be no other way when humans and social organizations of humans through history are involved in the Anthropocene topic. Regarding the Anthropocene , geosciences are not any more just a geologic or stratigraphic issue but a social issue too.

This statement simply demonstrates that the reviewer is unaware of efforts undertaken within geosciences during recent years (see about 50 references in ‘Science Direct’ or about 700 in ‘Google Scholar’).

So what? The Bible is the most cited text in history but I do not think anybody would take it as scientifically sounded.

Notwithstanding that one may question the work of other scholars, in this case that of Jonas, it is quite a ‘theological view of the world’ to claim that there is only ‘one road to Rome’.

As stated above, it is author’s duty to demonstrate that his way to Rome can effectively go to Rome and not to other places. That is, the author has to demonstrate that his idealist-based and ten commandments-style approach, which does not confront any of the material and socioeconomic reasons underlying the Anthropocene planetary crisis is of any use to face such crisis.

The reviewer may prefer a line of analysis as outlined in the remark; however, this essay is not about developing ethical concepts, but to use a tool, the ‘geoethical promise’, that is part of current geoethical thinking. The tool may have its limitations but has the advantage, for the given subject and audience, that it was developed within geosciences, and is gaining increasingly widespread acceptance in the geoscience community.

See above comments. My point is that such a tool, as it is conceived, is intrinsically useless.

Furthermore, the reviewer’s comments are further evidence of lack of knowledge of existing scholarly work on geoethics (appropriately referenced in my essay) and once again betray a dogmatic and unjustified insistence that there is a single acceptable path (apparently founded in Marxist theory) for research in this area.

See above comments about lots of non-Marxists papers on the Anthropocene issue that have contributed to build up a scientifically sounded and rigorous understanding of the Anthropocene. This is not the case of the present ms.

In my view the author has still time to produce a better manuscript for this special issue whose deadline is 31 July. Given that discussion between the author and me is at some stationary point in which ‘I say this you say that’, I decline any invitation to further review this manuscript in case there is a new revised submission.
